# Work participation and risk factors for health-related job loss among older workers in the Health and Employment after Fifty (HEAF) study: Evidence from a 2-year follow-up period

Holly E. Syddall[1,2]*, Stefania D'Angelo[1,2], Georgia Ntani[1,2], Martin Stevens[1,2], E. Clare Harris[1,2], Catherine H. Linaker[1,2], Karen Walker-Bone[1,2]

**1** Medical Research Council Lifecourse Epidemiology Unit, University of Southampton, Southampton, United Kingdom, **2** Medical Research Council Versus Arthritis Centre for Musculoskeletal Health and Work, Medical Research Council Lifecourse Epidemiology Unit, University of Southampton, Southampton, United Kingdom

* hes@mrc.soton.ac.uk

## Abstract

### Introduction

Rapidly increasing population old age dependency ratios create a growing economic imperative for people to work to older ages. However, rates of older worker employment are only increasing slowly. Amongst a cohort of contemporary older workers, we investigated risk factors for health-related job loss (HRJL) over 2 years of follow-up.

### Methods

HEAF is a population based cohort study of adults in England (aged 50–64 years at baseline) who provided information about socio-demographic characteristics, lifestyle, and work at baseline and annual follow-ups. Exits from paid work were mapped and risk factors for HRJL explored in a multiple-record survival dataset by Cox proportional hazards models.

### Results

2475 (75%) men and 2668 (66%) women were employed; 115 (4.6%) men and 182 (6.8%) women reported HRJL. Employment as road transport drivers/in vehicle trades (men), or as teaching/education/nursing/midwifery professionals or in caring personal services (women), was more frequent among people exiting work for health-related versus non-health-related reasons. Principal socio-demographic and lifestyle risk factors for HRJL were: struggling financially (men and women); low physical activity (men); being overweight or obese, and current smoking (women). Mutually adjusted work-related risk factors for HRJL were job dissatisfaction, and not coping with the physical (hazard ratio [95% confidence interval]: men 5.34[3.40,8.39]; women 3.73[2.48,5.60]) or mental demands (women only, 2.02[1.38,2.96]) of work.

**Data Availability Statement:** The data are not freely available owing to data protection and

consent restrictions, but may be accessed by collaboration with the HEAF study team. Enquiries should be directed to the HEAF principal investigator in the first instance (Prof Karen Walker-Bone, kwb@mrc.soton.ac.uk, Director of the MRC Versus Arthritis Centre for Musculoskeletal Health and Work, MRC Lifecourse Epidemiology Unit, University of Southampton) or the MRC Versus Arthritis Centre for Musculoskeletal Health and Work, MRC Lifecourse Epidemiology Unit (contact Mrs Sue Curtis, centre administrator, sc@mrc.soton.ac.uk, https://www.mrc.soton.ac.uk/cmhw/contact-us/).

**Funding:** KWB The HEAF study is funded by grant awards from Versus Arthritis (formerly Arthritis Research UK) (19817 and 20665) and the Medical Research Council programme grant (MC_UU_12011/5); and the Economic and Social Research Council and Medical Research Council jointly (ES/L002663/1); the study is coordinated by the MRC Lifecourse Epidemiology Unit, Southampton. https://www.versusarthritis.org https://mrc.ukri.org/ https://esrc.ukri.org/ The funders did not play any role in the study design, data collection and analysis, decision to publish, or preparation of the manuscript.

**Competing interests:** The authors have declared that no competing interests exist.

## Conclusions

Employment characteristics of contemporary older workers differ by sex. Job satisfaction and perceived ability to cope with the physical and mental demands of work are key determinants of HRJL which employers could potentially influence to enable work to older ages.

## Introduction

The populations of Western countries are ageing. In Europe, the number of people of working age (15–64 years) for each person aged ≥65 years is projected to decrease from over three in 2016 to under two by 2080 (an increase in the old-age dependency ratio from 29.3% to 52.3%) [1]. In a bid to mitigate the economic challenges posed by an ageing population and the projected insufficiency of resources for pensions, governments are introducing policies and legislation to encourage people to remain in paid work to older ages.

Macro level measures to extend working life may however be limited in their efficacy and run the risk of widening social inequalities and the disability employment gap because they do not recognise the complex individual level barriers and facilitators that accumulate throughout the lifecourse to affect a person's ability to extend their working life [2–4]. In particular, the capability of women to work to older ages is an especially pertinent issue in the light of the recent harmonisation of the pension age for men and women in many European countries. Government policies do not reflect that work is different for men and women across the lifecourse and that, in general, women's work is more likely to be insecure, part-time, poorly remunerated, and more likely to be dovetailed with family and caring responsibilities [3, 5]. Moreover, although many people thrive on work and will continue to reap its physical, mental, social and financial benefits at older ages, it is not universally feasible to maintain all types of occupations well into the seventh or eighth decades of life and this capability varies not only individually but also regionally and between communities [6]. Different occupations are associated with variable levels of physical and/or mental strain on employees and what is acceptable without deleterious consequence to health or wellbeing will change throughout the working lifecourse. Low-paid older workers who are employed in manual occupations, or who experience disability or chronic morbidity, may face the double jeopardy of needing to earn for longer but being unable to remain in their established job [6].

It has been suggested that strategies to extend working life should be individually-focussed, recognising the heterogeneity of older workers, and enabling them to continue to work in a way that matches their capabilities and needs, whilst complementing their wider social context [7–9]. Health is a major reason for early exit from employment [10]; a focus on identification of the wider drivers of health-related job loss (HRJL), irrespective of specific illnesses or diagnoses, therefore offers the potential to inform the development and implementation of workplace strategies to encourage and enable work to older ages.

We used OVID to search Medline and Embase for papers describing predictors of HRJL in general population samples of older workers. A combination of free-text terms were used to identify HRJL because disability retirement is not a formal exit mechanism from employment in all countries (the UK amongst them [11]) and relevant articles might otherwise have been missed. Risk factors identified included: poor socioeconomic circumstances [2, 12–16], low educational attainment [2, 12, 14–16], and difficult financial circumstances [14, 17]; lifestyle risk factors and poor health behaviours (smoking [18–25], high alcohol intake [19, 20], low physical activity [22, 24, 26–28] and obesity [16, 18, 20, 27, 29, 30]); physical demands of work

and ability to cope with them [13, 18, 24, 31–34]; mental job demands [32]; and psychosocial aspects of work including effort-reward imbalance, job demand and control [16, 24, 28, 33–38], and job satisfaction [32–34].

Only one study of 14,708 Dutch employees followed-up between 1999–2008 considered a wide panel of personal and employment related risk factors [24]; educational inequalities in health-related exit from work were shown to be partly mediated by health, lifestyle and work characteristics. Most of the studies cited above were from Scandinavia or other areas of mainland Europe, only two papers included data from the UK [2, 19]. Carr et al showed that workers with low socioeconomic position were at increased risk of HRJL at older ages in seven cohort studies (of which four were British); however, the only participant characteristics considered were age, sex, education, occupational grade, and self-rated health [2]. Hagger-Johnson et al showed that smoking, heavy drinking or a poor diet in midlife were risk factors for HRJL by early old age in 7,704 men and women in the Whitehall II study of civil servants; they called for future research into the mechanisms underlying their findings, and for consideration of gender differences [19].

To address this gap in knowledge we examined a broad panel of personal and employment-related risk factors for health-related job loss over two years of follow-up among participants in the Health and Employment After Fifty (HEAF) Study, UK.

## Methods

### Population

As described previously, the HEAF study follows a large population-based cohort of adults in England (aged 50–64 years at baseline) [39]. In brief, postal questionnaires were mailed to 39,359 adults aged 50–64 years identified through 24 general practices, drawn from every region of England and all deciles of social deprivation. Ethical approval was obtained from the NHS Research Ethics Committee North West-Liverpool East (Ref: 12/NW/0500). When they returned their baseline questionnaire, all participants gave written informed consent to participate in the study and to be sent annual follow-up questionnaires.

### Questionnaire

The baseline questionnaire enquired about: socio-demographics; lifestyle; employment status; and for those in paid work, its nature and perceptions about working conditions. This paper will explore whether the following characteristics are risk factors for HRJL during two years of follow-up: age; proximity to retirement; marital status; highest educational qualification; proportion of household income earned; financial dependents; housing tenure; self-perceived difficulty managing financially; receipt of private pension; physical activity (weekly hours); BMI (from self-reported height and weight); alcohol consumption (units per week categorised as: 'non-drinker or ≤1 unit'; '2–14 units'; '≥15 units'); smoking status (never/ex/current); contact with social network outside home; type of employment contract; duration of employment; size of workforce; working rotating or night shifts; physical work demands; perceived job security; job satisfaction; coping with the mental and physical demands of the job; and sick pay entitlement. Questionnaire response categories and groupings are detailed in S1 Appendix.

At baseline and each annual follow-up, participants were asked whether their employment had changed. If relevant, participants reported the dates of leaving and starting a job in the intervening period and stated whether a health problem was mainly or partly the reason for leaving work (referred to herein as a 'health-related job loss'). Participants who changed job were asked the same questions about their current employment as they had been asked about their previous employment. Those remaining in the same job updated their perception of how

well they were coping with its physical and mental demands. All follow-up respondents provided updated information on their marital status and financial circumstances.

## Statistical methods

Men and women were analysed separately throughout; this analysis strategy was decided *a priori* because work and its social context typically differ between men and women [3, 5, 40], and previous research has called for investigation of gender differences in risk factors for health-related job loss [19]. Analyses were conducted using the Stata statistical software package (release 15). Participant characteristics were summarised using frequency and percentage distributions, means and standard deviations, and medians and inter-quartile ranges.

The structure of each follow-up questionnaire enabled respondents to detail the date of leaving one job, and the date of starting a new job, in the time between subsequent HEAF questionnaires; accordingly, participants could report a maximum of two job exits between HEAF baseline and 2-year follow-up. Venn diagrams were used to describe the occurrence of job exits at the person-level during the 2-year period of follow-up for reasons: owing to health; not due to health; or unspecified. Overall person-level experience of job exit during 2-year follow-up was categorised as follows for descriptive purposes: no job exits; health-related job exit (s), with or without a job exit for other reasons; non-health-related job exit(s); job exit(s) for unspecified reasons only. 5,143 (70%) of the 7,303 HEAF participants who responded to follow-up 1 and/or 2 reported being in paid employment at some point between HEAF baseline and their follow-up(s) and comprised the group among whom employment patterns and job exits were described. Differences in the baseline characteristics of HEAF respondents according to sex and whether or not they were in paid employment between baseline and 2-year follow-up were examined by cross-tabulations and chi-squared tests.

To examine risk factors for HRJL we created a multiple-record, multiple-failure survival dataset, with time varying covariates for characteristics that potentially changed over time, such as financial circumstances, self-reported health, coping with the demands of work, and details about new jobs [41]. Each line of this dataset represented a period of time during which a respondent was 'at risk' of a health-related job loss (either: the time between two questionnaires during which employment status was unaltered; the time between a questionnaire and a job exit; or the time between the start of a job and the subsequent questionnaire). Each line of the dataset recorded the status of the respondent at the end of the time period as: in work; not in work for a health-related reason; not in work for a reason other than health; not in work for an unspecified reason. 5,032 HEAF participants provided sufficient information about dates of employment during the 2-year follow-up period to enable their inclusion in this survival dataset.

Our principal analyses used a multiple-record Cox proportional hazards model to explore risk factors for time to first HRJL event; in common with previous studies, we regarded other work outcomes (remaining in employment or job exits for other reasons) as censoring events [2]. We used Cox models to estimate hazard ratios (HR) and 95% confidence intervals (95% CI) for the relationship between participant characteristics and risk of HRJL using a complete case analysis approach. We adopted a forward selection modelling strategy with the aim of identifying key risk factors for HRJL within each of the domains of socio-demographic, lifestyle, and employment characteristics before moving on to final mutually adjusted models. First stage analyses estimated hazard ratios for HRJL in relation to one risk factor at a time, adjusted only for age. Second stage analyses focussed on risk factors that were significant at the 5% level ($p < 0.05$) in stage 1; mutually adjusted models within the domains of socio-demographic characteristics, lifestyle, and employment, were used to identify key risk factors for

HRJL in each domain. A final model estimated mutually adjusted hazard ratios for all of the socio-demographic, lifestyle, and employment characteristics that were significant in stage 2. Sensitivity analyses explored whether results were different for job loss which was 'mainly' as opposed to 'partly' due to health, and if the Anderson and Gill model was used to analyse time to any HRJL (multiple failures included). Tests of the proportional hazards assumption in the final models were based on Schoenfeld residuals and implemented using the *estat phtest* command in Stata. Log-log plots were also used to graphically assess the proportional-hazards assumption.

### Study sample

In all, 8,134 participants completed a baseline HEAF questionnaire in 2013/2014. 7,303 (90%) of these responded to at least one of the two annual follow-ups, amongst whom 5,143 (70%) were in paid employment at some point and comprised the sample for descriptive analyses. 5,032 of these provided sufficient information for inclusion in the survival analyses.

## Results

### Characteristics of HEAF participants

Table 1 describes the participants by sex and employment status between baseline and 2-year follow-up; 75% of men and 66% of women were employed at some point. Table 1 shows that people who were employed at some point were on average more likely to be: younger; to have financial dependents; to mortgage rather than own their home outright; and to be doing some weekly physical activity. Men who worked were also more likely to be never smokers (although the proportion of current smokers was similar), and were more likely to be in the overweight BMI category (although the proportion who were obese was similar). Employed men were more likely to be married, and employed women were less likely to be married but more likely to be better educated, than their non-working counterparts.

Table 1 also reveals sex differences in the characteristics of employed HEAF participants. With respect to socio-demographic and lifestyle characteristics, men were, on average, more likely than women to: be married; have qualifications higher than school level; earn ≥50% of household income; have financial dependents; be financially comfortable; have access to a private pension; have no weekly contact with friends/family outside the household; be in the overweight BMI category; be heavy drinkers or ex smokers. In terms of their work, men were more likely than women to: be self-employed; work nights; have a physically-demanding job; have a very short or very long entitlement to sick pay; and be eligible for an ill-health retirement pension. In contrast, women were more likely than men to report difficulties coping with the mental demands of their job.

### Employment exits

603 (24.4%) men and 687 (25.8%) women reported leaving paid employment between baseline and 2-year follow-up. Of these, 324 (53.7%) men and 383 (55.7%) women exited employment with no subsequent return to work; of those subsequently re-employed, 43 (15.4%) men and 51 (16.8%) women also left those jobs. S2 Appendix shows the distribution of HEAF participants who left a job by sex and reason for exit.

### Health-related job exits

115 men and 182 women reported leaving a job between baseline and 2-year follow-up because of their health (4.6% and 6.8% of employed men and women), with two of the men and eight

**Table 1. Baseline characteristics according to sex and employment status between HEAF baseline and 2-year follow-up.**

| N(%) | MEN | | WOMEN | |
|---|---|---|---|---|
| | **No work** | **Any work** | **No work** | **Any work** |
| | **(n = 804)** | **(n = 2475)** | **(n = 1356)** | **(n = 2668)** |
| *Socio-demographic* | | | | |
| **Age at baseline (years)[+]** | 61.9 (3.6) | 57.8 (4.2) | 61.7 (3.6) | 57.2 (3.9) |
| *Proximity to expected retirement* | | | | |
| <1 year | n/a | 134 (5.7) | n/a | 153 (6.2) |
| 1 to <5 years | | 585 (25.0) | | 537 (21.9) |
| 5 to <10 years | | 779 (33.3) | | 916 (37.4) |
| 10 years or more | | 843 (36.0) | | 845 (34.5) |
| *Marital status* | | | | |
| Married/civil partnership | 559 (69.7) | 1907 (77.2) | 988 (73.4) | 1754 (66.5) |
| Single/widowed/divorced | 243 (30.3) | 563 (22.8) | 358 (26.6) | 885 (33.5) |
| *Highest educational qualification* | | | | |
| No qualifications/school | 262 (32.6) | 737 (29.8) | 593 (43.7) | 952 (35.7) |
| Vocational training certificate | 232 (28.9) | 823 (33.3) | 335 (24.7) | 804 (30.1) |
| University degree/higher | 310 (38.6) | 915 (37.0) | 428 (31.6) | 912 (34.2) |
| *Proportion of family income earned by you* | | | | |
| None | 680 (89.8) | 82 (3.4) | 1181 (93.6) | 130 (5.0) |
| Less than a quarter | 22 (2.9) | 104 (4.3) | 36 (2.9) | 415 (16.1) |
| Between a quarter and a half | 14 (1.8) | 336 (13.9) | 13 (1.0) | 652 (25.3) |
| Half or more | 41 (5.4) | 1893 (78.4) | 32 (2.5) | 1385 (53.6) |
| *Financial dependents outside your household* | | | | |
| No | 753 (95.0) | 2181 (89.9) | 1280 (96.2) | 2395 (91.9) |
| Yes | 40 (5.0) | 246 (10.1) | 51 (3.8) | 211 (8.1) |
| *Housing tenure* | | | | |
| Owned outright | 553 (69.5) | 1162 (47.6) | 992 (74.0) | 1288 (49.4) |
| Mortgaged | 93 (11.7) | 990 (40.6) | 160 (11.9) | 1003 (38.5) |
| Rented/rent free | 150 (18.8) | 287 (11.8) | 188 (14.0) | 314 (12.1) |
| *How are you managing financially?* | | | | |
| Living comfortably/doing alright | 556 (69.8) | 1759 (72.1) | 997 (74.5) | 1809 (69.3) |
| Just about getting by | 156 (19.6) | 501 (20.5) | 229 (17.1) | 554 (21.2) |
| Finding it difficult/very difficult | 84 (10.6) | 180 (7.4) | 113 (8.4) | 246 (9.4) |
| *Access to private pension* | | | | |
| State pension only | 94 (11.8) | 300 (12.2) | 392 (29.4) | 613 (23.2) |
| Private pension now/future | 703 (88.2) | 2155 (87.8) | 943 (70.6) | 2024 (76.8) |
| *Lifestyle* | | | | |
| *Weekly physical activity* | | | | |
| Some | 588 (79.9) | 1910 (84.0) | 910 (78.0) | 1935 (82.0) |
| None | 148 (20.1) | 363 (16.0) | 256 (22.0) | 425 (18.0) |
| *Weekly contact with friends/family not in your household* | | | | |
| Some | 661 (89.8) | 2016 (89.3) | 1231 (96.1) | 2414 (94.9) |
| None | 75 (10.2) | 242 (10.7) | 50 (3.9) | 129 (5.1) |
| *Obesity* | | | | |
| Normal/underweight <25kg/m² | 260 (33.2) | 663 (27.5) | 540 (41.1) | 1140 (44.0) |
| Overweight 25–29.9kg/m² | 335 (42.8) | 1179 (48.9) | 448 (34.1) | 834 (32.2) |
| Obese/severely obese ≥30kg/m² | 188 (24.0) | 570 (23.6) | 326 (24.8) | 616 (23.8) |
| *Alcohol intake per week* | | | | |

(*Continued*)

**Table 1.** (Continued)

| N(%) | MEN | | WOMEN | |
|---|---|---|---|---|
| | **No work** | **Any work** | **No work** | **Any work** |
| | **(n = 804)** | **(n = 2475)** | **(n = 1356)** | **(n = 2668)** |
| Low/no drinker (≤1unit pwk) | 119 (16.3) | 307 (13.1) | 342 (30.6) | 672 (28.2) |
| Moderate (2–14 units pwk) | 339 (46.4) | 1219 (52.2) | 685 (61.3) | 1523 (64.0) |
| Heavy (15+ units pwk) | 272 (37.3) | 810 (34.7) | 90 (8.1) | 185 (7.8) |
| *Smoking status* | | | | |
| Never | 352 (44.3) | 1263 (51.4) | 778 (58.1) | 1513 (57.3) |
| Ex | 350 (44.0) | 922 (37.5) | 437 (32.6) | 864 (32.7) |
| Current | 93 (11.7) | 271 (11.0) | 125 (9.3) | 262 (9.9) |
| *Employment*[++] | | | | |
| *Type of contract* | | | | |
| Permanent | | 1776 (74.1) | | 2156 (82.4) |
| Temporary/renewable | | 167 (7.0) | | 169 (6.5) |
| Self-employed | | 455 (19.0) | | 290 (11.1) |
| *Duration of current employment* | | | | |
| Less than 1 year | | 220 (9.2) | | 248 (9.4) |
| 1 to 5 years | | 442 (18.4) | | 435 (16.5) |
| More than 5 years | | 1737 (72.4) | | 1952 (74.1) |
| *Number of people who work for employer* | | | | |
| Just you | | 307 (12.9) | | 216 (8.3) |
| 2–9 | | 301 (12.7) | | 308 (11.9) |
| 10–29 | | 256 (10.8) | | 332 (12.8) |
| 30–499 | | 641 (27.0) | | 745 (28.8) |
| 500 or more | | 872 (36.7) | | 989 (38.2) |
| *Job involves rotating/variable shifts* | | | | |
| Sometimes/rarely/never | | 1999 (83.9) | | 2204 (84.7) |
| Often | | 384 (16.1) | | 398 (15.3) |
| *Job involves night work* | | | | |
| Sometimes/rarely/never | | 2195 (92.0) | | 2507 (96.2) |
| Often | | 190 (8.0) | | 98 (3.8) |
| *Physical work score*[+++] | | 1 (0,6) | | 0 (0,6) |
| *Job satisfaction* | | | | |
| Very satisfied/satisfied | | 2226 (92.6) | | 2456 (94.0) |
| Dissatisfied/very dissatisfied | | 179 (7.4) | | 158 (6.0) |
| *Job security* | | | | |
| Secure when well or ill | | 1220 (50.8) | | 1382 (52.8) |
| Insecure when well or ill | | 1183 (49.2) | | 1233 (47.2) |
| *Duration of sick pay* | | | | |
| Less than one week | | 506 (21.8) | | 401 (16.0) |
| 1 to 4 weeks | | 239 (10.3) | | 247 (9.8) |
| 1 to 6 months | | 923 (39.7) | | 1164 (46.3) |
| More than 6 months | | 231 (9.9) | | 142 (5.7) |
| Not sure | | 426 (18.3) | | 559 (22.2) |
| *Ill-health retirement pension entitlement* | | | | |
| No | | 1163 (48.8) | | 1142 (44.1) |
| Yes | | 596 (25.0) | | 529 (20.4) |
| Don't know | | 625 (26.2) | | 920 (35.5) |

*(Continued)*

**Table 1.** (Continued)

| N(%) | MEN | | WOMEN | |
|---|---|---|---|---|
| | **No work** | **Any work** | **No work** | **Any work** |
| | **(n = 804)** | **(n = 2475)** | **(n = 1356)** | **(n = 2668)** |
| *Currently coping with physical demands of the job* | | | | |
| Easily | | 1729 (71.9) | | 1829 (70.0) |
| Some difficulty or more | | 676 (28.1) | | 784 (30.0) |
| *Currently coping with mental demands of the job* | | | | |
| Easily | | 1705 (71.0) | | 1748 (66.9) |
| Some difficulty or more | | 696 (29.0) | | 865 (33.1) |

n/a: not applicable; pwk: per week.

Statistics are frequency and percentage distributions within sex and worker status groups.

[+]Mean and standard deviation.

[++]For descriptive purposes, in this table only, employment characteristics were coded from the first job reported between HEAF baseline and 2-year follow-up; this was at baseline for 97% (2,399 men and 2,577 women) of the sample, at 1 year follow-up for 2% (50 men and 46 women), and at 2-year follow-up for 1% (26 men, 45 women).

[+++]Median and inter-quartile range.

P<0.05 for differences in baseline characteristics by work status within men, and within women, for all characteristics except for access to private pension and social network in men, and social network, obesity, alcohol intake and smoking in women.

P<0.05 for sex difference among workers for the following baseline characteristics: age, proximity to expected retirement, marital status, educational qualifications, proportion of family income earned by the individual, financial dependents, how managing financially, access to a private pension, weekly social contact, obesity, alcohol intake, smoking; type of employment contract, number of people working for employer, night work, duration of sick pay entitlement, ill health pension entitlement, physical work score, difficulty coping with mental demands of the job.

P-values estimated by: ANOVA for age; Mann-Whitney ranksum test for physical work score; and chi-squared tests for all other characteristics.

of the women reporting two health-related job exits (S2 Appendix). Of those reporting a HRJL, 49 (42.6%) men and 69 (37.9%) women reported that health was mainly, rather than partly, the reason for leaving their employment. When asked to attribute their health-related exit, 44 (38.3%) men and 72 (39.6%) women indicated a musculoskeletal problem; 34 (29.6%) men and 70 (38.5%) women indicated a mental health problem; 16 (13.9%) men and 16 (8.8%) women indicated a heart or lung problem; and 36 (31.3%) men and 70 (38.5%) women indicated an 'other' health problem (more than one health problem could be attributed).

## Occupations by work pattern between baseline and 2-year follow-up

Fig 1 shows the percentage distribution of prevailing occupation (coded to 1-digit level of SOC2010) according to sex and work pattern over the 2-year period of follow-up. Sex differences are apparent; on the whole, men were more likely than women to be employed in skilled trades, in process, plant or machine operative roles, or in elementary occupations. In contrast, women were more likely than men to be employed in administrative and secretarial occupations, caring, leisure and other service roles, or in sales and customer service.

Fig 1 also shows that men reporting HRJL were less likely to be employed as managers, directors and senior officials, in professional occupations, or in associate professional and technical occupations, than those who stopped working not for health reasons.

S3A and S3B Appendices expand on Fig 1 and show the distribution of prevailing occupational codes at the 3-digit level of SOC2010 by work pattern. The most frequently occurring individual occupations among men who left work for health-related reasons were 'Road transport drivers' and 'Vehicle Trades'; for women these were 'Teaching and Educational professionals', 'Nursing and Midwifery professionals' and 'Caring Personal Services'.

## Men

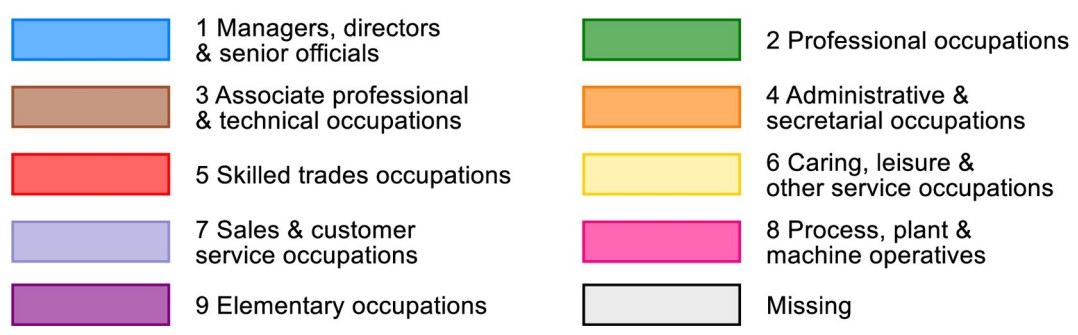

## Women

1 Managers, directors & senior officials

2 Professional occupations

3 Associate professional & technical occupations

4 Administrative & secretarial occupations

5 Skilled trades occupations

6 Caring, leisure & other service occupations

7 Sales & customer service occupations

8 Process, plant & machine operatives

9 Elementary occupations

Missing

**Fig 1. SOC2010 1-digit prevailing* job code by work pattern between HEAF baseline and 2-year follow-up.** *Job codes are those prevailing at the time of the first job exit of the type indicated. Job code for first reported job between baseline and 2-year follow-up is graphed for people in work with no exits. HRJL: health-related job loss.

### Longitudinal analysis of risk factors for health-related job loss

The 2-year survival analysis file included 2418 men and 2614 women of whom 108 and 176 experienced a HRJL respectively. Rates of HRJL per 1,000 person-years employed were 25.0 (95%CI 20.7, 30.2) for men and 38.3 (95%CI 33.0, 44.4) for women. Table 2 shows hazard ratios for HRJL for one risk factor at a time, adjusted for age. Characteristics associated with increased risk of HRJL among men and women were: close proximity to expected retirement; difficulty managing financially; no weekly physical activity; job dissatisfaction; job insecurity; and difficulty coping with the physical, and mental, demands of the job. In addition, owning one's home outright, self-employment and lower physical work score were associated with reduced likelihood of HRJL among men. Women who were highly educated, were obese/ severely obese, or were current smokers, were at increased risk of HRJL. No other job characteristics were associated with HRJL.

In final mutually-adjusted models (Table 3), the important socio-demographic risk factors for HRJL were: close proximity to retirement (both sexes), and high educational level (women only). Important lifestyle risk factors were: no weekly physical activity among men, and current smoking and being in the overweight BMI category among women. Job dissatisfaction and difficulty coping with the physical demands of the job were employment related risk factors among men and women (Table 3 and Fig 2). However, difficulty coping with the mental demands of the job was only a risk factor among women. Occurrence of HRJL varied markedly according to number of employment related risk factors: for example, HRJL was only experienced by 2.0% of men and 3.4% of women who reported being satisfied with their job and coping with its physical and mental demands. In contrast, HRJL was experienced by 21.9% of men and 24.5% of women who were dissatisfied with their job, and not coping with its physical or mental demands.

## Discussion

In this contemporary prospective cohort study, we explored the sectors in which older workers in England are employed, described the characteristics of their employment and exits from it, and evaluated a wide range of potential risk factors for self-reported health-related job loss. A quarter of workers exited paid work during the two years of follow-up, with a quarter of them doing so for health reasons; women reported HRJL more frequently than men. As expected, the predominant occupations of male and female older workers differed, and the occupations exited relatively more frequently for health than non-health reasons also differed (these were: road transport drivers and vehicle trades occupations (men), and educational, health and social care occupations (women)). Job dissatisfaction and difficulty coping with the physical demands of work were the dominant risk factors for HRJL, independent of socio-demographic and lifestyle factors. Difficulty coping with the mental demands of work was an additional risk factor for women. Overall, HRJL was reported by 2.0% of men and 3.4% of women who were satisfied with their job and coping with its physical and mental demands, but by 21.9% of men and 24.5% of women who were dissatisfied and not coping with these demands.

Our work considers self-reported exit from employment mainly or partly for health reasons and we suggest that it is particularly valuable in doing so. In some countries, people who retire early on health grounds are eligible for a disability pension and are classified as having a "disability retirement"; indeed this type of exit from employment constituted the outcome variable

**Table 2. Risk factors for health-related job loss: one risk factor at a time, adjusted for age.**

|  | Age-adjusted HR (95%CI) | |
|---|---|---|
|  | Men | Women |
| *Socio-demographic* |  |  |
| **Age in years** | 1.08 (1.03,1.13) | 1.08 (1.04,1.13) |
| **Proximity to expected retirement** |  |  |
| <1 year | 5.99 (2.79,12.88) | 5.34 (3.14,9.11) |
| 1 to <5 years | 2.39 (1.38,4.16) | 1.26 (0.82,1.94) |
| 5 to <10 years | Ref | Ref |
| 10 years or more | 0.98 (0.53,1.79) | 0.72 (0.46,1.13) |
| *Marital status* |  |  |
| Married/civil partnership | 0.81 (0.53,1.25) | 0.77 (0.57,1.04) |
| Single/widowed/divorced | Ref | Ref |
| *Highest educational qualification* |  |  |
| No qualifications/school | 1.14 (0.72,1.79) | 0.69 (0.48,0.99) |
| Vocational training certificate | 0.97 (0.61,1.55) | 0.83 (0.58,1.18) |
| University degree/higher | Ref | Ref |
| *Proportion of family income earned by you* |  |  |
| None | 1.98 (0.72,5.40) | 0.63 (0.23,1.72) |
| Less than a quarter | 0.85 (0.31,2.32) | 0.64 (0.39,1.03) |
| Between a quarter and a half | 0.83 (0.46,1.49) | 0.94 (0.66,1.33) |
| Half or more | Ref | Ref |
| *Financial dependents outside your household* |  |  |
| No | Ref | Ref |
| Yes | 1.05 (0.56,1.97) | 1.22 (0.73,2.04) |
| *Housing tenure* |  |  |
| Owned outright | Ref | Ref |
| Mortgaged | 0.77 (0.49,1.20) | 0.69 (0.48,0.98) |
| Rented/rent free | 1.78 (1.07,2.95) | 1.16 (0.75,1.79) |
| *How are you managing financially?* |  |  |
| Comfortable/doing alright | Ref | Ref |
| Getting by | 1.06 (0.65,1.72) | 1.50 (1.06,2.14) |
| Very/difficult | 1.84 (1.00,3.40) | 2.27 (1.46,3.53) |
| *Access to private pension* |  |  |
| State pension only | 1.05 (0.59,1.88) | 1.08 (0.77,1.53) |
| Private pension now/future | Ref | Ref |
| *Lifestyle* |  |  |
| *Weekly physical activity* |  |  |
| Some | Ref | Ref |
| None | 2.42 (1.59,3.67) | 1.53 (1.06,2.21) |
| *Weekly contact with friends/family not in your household* |  |  |
| Some | Ref | Ref |
| None | 1.37 (0.78,2.41) | 1.48 (0.82,2.67) |
| *Obesity* |  |  |
| Normal/underweight <25kg/m$^2$ | Ref | Ref |
| Overweight 25–29.9kg/m$^2$ | 1.14 (0.71, 1.83) | 1.39 (0.98, 1.99) |
| Obese/severely obese ≥30kg/m$^2$ | 1.22 (0.71, 2.08) | 1.54 (1.06, 2.24) |
| *Alcohol intake per week* |  |  |
| Low/no drinker (≤1unit pwk) | Ref | Ref |

(*Continued*)

**Table 2.** (Continued)

| | Age-adjusted HR (95%CI) | |
|---|---|---|
| | **Men** | **Women** |
| Moderate (2–14 units pwk) | 0.71 (0.42,1.22) | 0.69 (0.49,0.97) |
| Heavy (15+ units pwk) | 0.72 (0.41,1.27) | 0.86 (0.47,1.58) |
| *Smoking status* | | |
| Never | Ref | Ref |
| Ex | 1.29 (0.86, 1.95) | 1.36 (0.98,1.88) |
| Current | 1.35 (0.75, 2.46) | 2.12 (1.38,3.25) |
| *Employment* | | |
| *Employment contract* | | |
| Permanent | Ref | Ref |
| Temporary/renewable | 1.44 (0.76,2.71) | 1.17 (0.66,2.06) |
| Self-employed | 0.50 (0.27,0.92) | 0.57 (0.32,1.03) |
| *Duration of current employment* | | |
| Less than 1 year | 1.28 (0.68,2.41) | 1.09 (0.64,1.86) |
| 1 to 5 years | 1.08 (0.67,1.74) | 1.27 (0.87,1.84) |
| More than 5 years | Ref | Ref |
| *Number of people who work for employer* | | |
| Just you | 0.38 (0.18,0.80) | 0.60 (0.32,1.13) |
| 2–9 | 0.64 (0.34,1.21) | 0.41 (0.22,0.76) |
| 10–29 | 0.51 (0.24,1.08) | 0.52 (0.29,0.92) |
| 30–499 | 0.74 (0.47,1.17) | 0.90 (0.64,1.27) |
| 500 or more | Ref | Ref |
| *Job involves rotating/variable shifts* | | |
| Sometimes/rarely/never | Ref | Ref |
| Often | 1.24 (0.76,2.04) | 1.16 (0.79,1.72) |
| *Job involves night work* | | |
| Sometimes/rarely/never | Ref | Ref |
| Often | 1.16 (0.58,2.29) | 1.26 (0.62,2.57) |
| *Physical work score* | 1.13 (1.00,1.24) | 1.09 (0.98,1.23) |
| *Job satisfaction* | | |
| Very satisfied/satisfied | Ref | Ref |
| Dissatisfied/very dissatisfied | 4.72 (2.99,7.45) | 4.07 (2.76,6.00) |
| *Job security* | | |
| Secure when well or ill | Ref | Ref |
| Insecure when well or ill | 2.12 (1.42,3.17) | 2.03 (1.50,2.76) |
| *Duration of sick pay* | | |
| Less than one week | 1.11 (0.68,1.82) | 0.80 (0.51,1.26) |
| 1 to 4 weeks | 1.02 (0.52,1.99) | 0.99 (0.59,1.64) |
| 1 to 6 months | Ref | Ref |
| More than 6 months | 0.81 (0.38,1.74) | 0.71 (0.35,1.48) |
| Not sure | 0.96 (0.56,1.64) | 0.68 (0.44,1.04) |
| *Ill-health retirement pension entitlement* | | |
| No | 0.91 (0.55,1.50) | 0.88 (0.58,1.34) |
| Yes | Ref | Ref |
| Don't know | 1.38 (0.82,2.33) | 1.32 (0.88,1.99) |
| *Currently coping with physical demands of the job* | | |
| Easily | Ref | Ref |

(*Continued*)

**Table 2.** (Continued)

|  | Age-adjusted HR (95%CI) | |
| --- | --- | --- |
|  | **Men** | **Women** |
| Some difficulty or more | 5.74 (3.82,8.65) | 5.60 (4.04,7.77) |
| *Currently coping with mental demands of the job* |  |  |
| Easily | Ref | Ref |
| Some difficulty or more | 2.47 (1.66,3.67) | 4.27 (3.12,5.85) |

Ref: reference category; pwk: per week; HR (95%CI): hazard ratio (95% confidence interval).

Results are based on the 2-year longitudinal survival analysis file containing information for 2418 men and 2614 women (108 and 176 of whom experienced a health-related job loss respectively).

All sociodemographic and lifestyle characteristics were analysed as fixed baseline covariates with the exception of managing financially, which was modelled as a time-varying covariate in common with all employment characteristics.

in many of the studies that we identified in our literature review of risk factors for health-related job loss. However, "disability retirement" may only be the tip of the iceberg of health-related early exits from work, and it is not a formal mechanism of exit from employment in all countries (the UK amongst them [11]); disability pension provision in the UK is voluntary (financed privately by the individual or employer). This limits the generalisability of findings between different countries and it is also possible that different characteristics may be implicated as risk factors for a job loss that a person, rather than a set of state defined criteria, attributes to health reasons. People who would qualify for formal disability retirement or benefit can reasonably be expected to comprise a subset of those who self-report exit from work for health reasons. We feel that our consideration of a broad sample of people in whom health is implicated in their reason for stopping work, provides important insights for the development of interventional strategies to extend working life [2, 13].

It is no surprise that work differs in men and women [40] and yet macro level legislation such as changing the age of eligibility for a state pension takes no account of gender differences, or indeed any other heterogeneity in older workers. Our data pertain to a contemporary cohort of older workers in England who are progressing through their retirement transition at a time when recent legislation to raise and harmonise state pension age is taking effect: UK women born in the 1950s (HEAF participants were born between 1949 and 1963) have seen their age of eligibility for state pension rise from 60 to 66 years over one decade and further increases in state pension age are scheduled. HEAF women who reported HRJL during follow-up were particularly employed in educational, health and social care occupations. Employers can play a fundamental role in encouraging fuller working lives by enabling older workers to match their work with their life [42]. Employer led initiatives to retain older workers include [42]: encouraging and enabling flexible working which might include part-time or remote working, or variable start and finish times, which complement the caring responsibilities, physical capabilities and long-term health conditions that comprise the life context of each individual employee [43, 44]; providing opportunities for reskilling and redeployment to less physically demanding roles if desired by the employee; and listening to, engaging with, and responding to the needs of older workers so they feel a valued part of the workforce with opportunities and benefits equal to those of their younger colleagues. The relative importance of these employer based strategies for retaining older workers is likely to differ for employers with a mixed, predominantly male, or predominantly female workforce. Given that older working women are more likely to also have caring responsibilities than their male

**Table 3. Mutually adjusted hazard ratios for health-related job loss by sex.**

| Risk factor | Men HRJL N | Men Person-years employed (1000's) | Men Mutually adjusted HR (95%CI) | Women HRJL N | Women Person-years employed (1000's) | Women Mutually adjusted HR (95%CI) |
|---|---|---|---|---|---|---|
| **Proximity to expected retirement** | | | | | | |
| Less than a year | 10 | 0.0907 | 5.72 (2.57,12.74) | 24 | 0.1134 | 7.60 (4.24,13.64) |
| 1 to <5 years | 37 | 0.8838 | 2.50 (1.43,4.37) | 39 | 0.8499 | 1.45 (0.91,2.30) |
| 5 to <10 years | 24 | 1.2645 | Ref | 51 | 1.5478 | Ref |
| 10+ years | 23 | 1.3974 | 0.98 (0.52,1.82) | 34 | 1.4595 | 0.77 (0.47,1.26) |
| **Highest educational qualification** | | | | | | |
| No qualification/school | | | | 46 | 1.4085 | 0.59 (0.39,0.87) |
| Vocational training certificate | | | | 42 | 1.2068 | 0.63 (0.42,0.95) |
| University degree/higher | | | | 60 | 1.3554 | Ref |
| **How are you managing financially?** | | | | | | |
| Comfortable/doing alright | 66 | 2.6725 | Ref | 88 | 2.7988 | Ref |
| Getting by | 17 | 0.7422 | 0.66 (0.38,1.14) | 40 | 0.8356 | 1.17 (0.80,1.73) |
| Very/difficult | 11 | 0.2216 | 1.14 (0.58,2.22) | 20 | 0.3362 | 1.32 (0.78,2.24) |
| **Weekly physical activity** | | | | | | |
| Some | 64 | 3.0593 | Ref | | | |
| None | 30 | 0.5771 | 2.38 (1.53,3.70) | | | |
| **Smoking status** | | | | | | |
| Never | | | | 72 | 2.3694 | Ref |
| Ex | | | | 53 | 1.2365 | 1.29 (0.90,1.84) |
| Current | | | | 23 | 0.3647 | 1.67 (1.02,2.74) |
| **Obesity** | | | | | | |
| Normal/underweight <25kg/m$^2$ | | | | 50 | 1.7277 | Ref |
| Overweight 25–29.9kg/m$^2$ | | | | 56 | 1.2705 | 1.51 (1.03,2.22) |
| Obese/severely obese ≥30kg/m$^2$ | | | | 42 | 0.9725 | 1.15 (0.76,1.75) |
| **Job satisfaction** | | | | | | |
| Very satisfied/satisfied | 74 | 3.4072 | Ref | 122 | 3.7550 | Ref |
| Dissatisfied/very dissatisfied | 20 | 0.2291 | 2.92 (1.73,4.93) | 26 | 0.2157 | 1.76 (1.12,2.77) |
| **Currently coping with physical demands of the job** | | | | | | |
| Easily | 30 | 2.6781 | Ref | 45 | 2.8346 | Ref |
| Some difficulties or more | 64 | 0.9583 | 5.34 (3.40,8.39) | 103 | 1.1361 | 3.73 (2.48,5.60) |
| **Currently coping with mental demands of the job** | | | | | | |
| Easily | | | | 55 | 2.7122 | Ref |
| Some difficulties or more | | | | 93 | 1.2584 | 2.02 (1.38,2.96) |

N: number; HRJL: health-related job loss; HR: hazard ratio; 95%CI: 95% confidence interval; Ref: reference category.

counterparts [3, 5], our study suggests that flexible working policies might be of particular importance for the prevention of HRJL in sectors with predominantly female workforces such as education, health and social care.

Relationships between socio-economic disadvantage and increased risk of health-related exit from work have been reported previously [2, 13]. The implication is a risk of widening social inequality as a consequence of universal policies and legislation to lengthen working life [2] if those from deprived backgrounds are unlikely to keep working until state pension age. HEAF men who rented rather than mortgaged or owned their home, or who were finding it

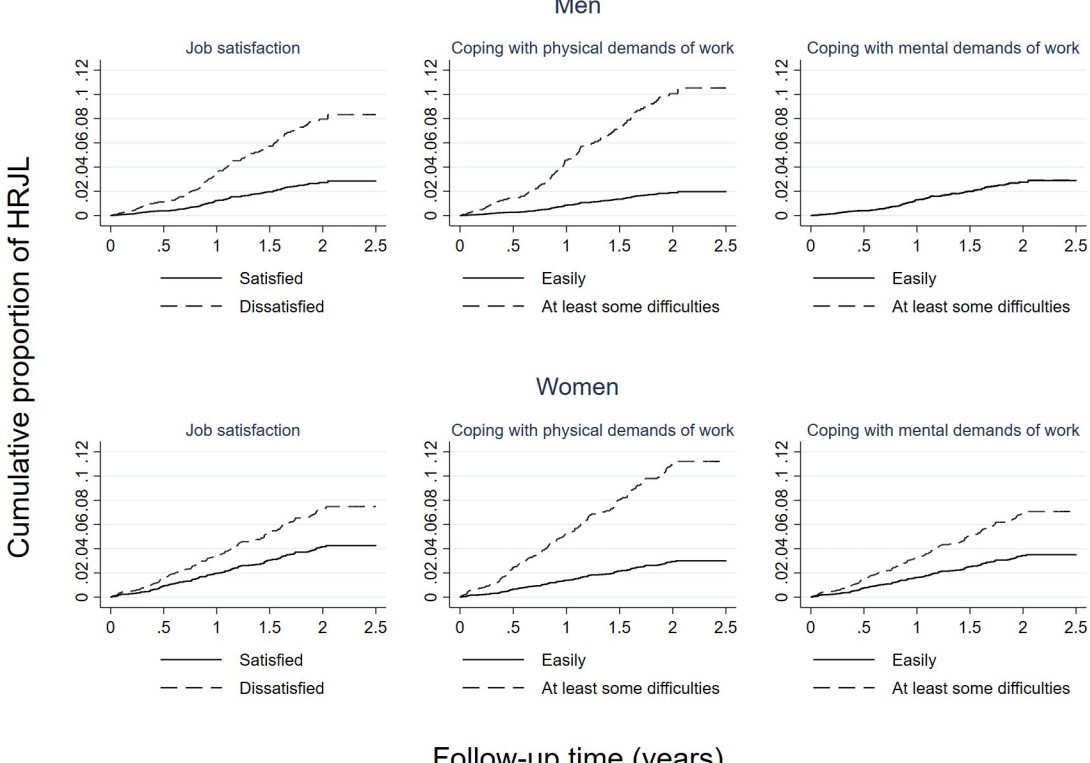

**Fig 2. Employment related risk factors for health-related job loss by sex.** HRJL: health-related job loss. Each plot was derived from the final mutually adjusted models for HRJL as presented in Table 3, with the exception of 'coping with the mental demands of work' among men; this plot was derived for illustrative purposes only from a model which included this work characteristic in addition to all of the variables included in the final model for men.

difficult to manage financially, were at increased risk of HRJL in univariate analyses. However, no socio-economic marker remained important in our fully adjusted model for men, suggesting that proximity to retirement, physical activity, job satisfaction and difficulty coping with the physical demands of work may mediate the impact of socio-economic position on HRJL among HEAF men. Our results suggest a more nuanced relationship between socio-economic position and HRJL among women. In univariate analyses, HEAF women who reported struggling financially were at increased risk of HRJL. However, so were highly educated women, and education remained important in the final mutually adjusted model. Further investigation revealed that incidence of HRJL was particularly high (16%) among women who were highly educated but struggling financially; incidence was 10% among women with low education who were struggling financially. Incidence of HRJL among women who were financially comfortable varied little by education (6% and 5% among women with high or low education respectively). The group of highly educated women who were struggling financially were also more likely than other women to: be divorced/single; earn ≥50% of their household income; have financial dependents; not own their home; work for a large employer, for a short amount of time, on a temporary contract; report job insecurity, dissatisfaction and difficulty coping with work's physical and mental demands. No such effects were evident among men. Interestingly, another study that considered a wide panel of personal and employment-related characteristics of 14,708 Dutch employees [24], found that the effect of education as a predictor of exit from paid employment through disability benefits was partly mediated by health, lifestyle and work characteristics. Our findings appear to support this. The patterns observed in our

study are consistent with the suggestion that the consequences of divorce for men are transient, but for women become chronic, resulting in long term loss of income and increased risk of poverty [45]. Our results also hint that these consequences may be particularly acute among women for whom lifetime expectation of socio-economic status (as reflected by education) is not matched by reality in later life. The interplay between educational level, financial circumstances, and marital status as predictors of HRJL merits further investigation.

Relationships between socio-economic position and HRJL are likely to be at least partly attributable to differences in lifestyle factors and health behaviours. Previous studies have shown that smoking [18–25], alcohol use [19, 20], physical inactivity [22, 24, 26–28], poor diet [19] and obesity [16, 18, 20, 27, 29, 30] increase the risk of exit from work through disability retirement or receipt of disability benefits. In our study, physical inactivity among men, and current smoking and overweight among women, were factors associated with HRJL in final models mutually adjusted for proximity to retirement, socio-economic position, job satisfaction and ability to cope with the demands of work. Diet was not assessed in the early phases of HEAF although a short food-frequency questionnaire has been included in more recent follow-ups. We asked about lifestyle risk factors using standard questionnaire tools but it is possible that these were not sensitive to identify a wider set of associations between lifestyle factors and HRJL. A healthy participant effect, common to most cohort studies [46], may have also influenced our results if it diminished the range of lifestyle exposures arising among the study participants and limited our ability to discriminate them as risk factors for HRJL; however, many lifestyle risk factors were broadly similar among those who did any work between baseline and follow-up and those who did none, lessening this concern.

Another way in which social factors might affect risk of HRJL is through the nature and demands of work. Men who were self-employed or whose work did not have heavy physical demands were less likely to report HRJL. Not coping with the physical demands of one's work was an important risk factor for HRJL in our final models for both men and women, robust to adjustment for all other factors and attenuating the association with difficulty managing financially. Other researchers have reported similar results in studies of exit from work through disability retirement, receipt of disability benefits, or voluntary early retirement [13, 18, 24, 31–34]. It seems that physically-demanding work (more common among people from more deprived backgrounds) becomes more difficult at older ages and that individuals who perceive a mismatch between what their work requires of them and their physical capacity are more likely to experience HRJL. The perceived mismatch may be explained by a growing burden of age-related comorbidities, for example osteoarthritis or COPD, that impact on functional capability, or it may be the perception of their capability relative to perceived demands that changes over time; this requires exploration in future research.

Job dissatisfaction was a dominant risk factor for HRJL among both men and women independent of socio-demographic and lifestyle factors, findings consistent with those from European studies which have shown similar associations between job dissatisfaction [32–34] and other poor psychosocial work characteristics (effort-reward imbalance and low job control) [16, 24, 28, 33–38] and exit from work through disability retirement, receipt of disability benefits, or voluntary early retirement. Job satisfaction is a complex multi-faceted phenomenon incorporating perceived aspects of the work and its rewards as compared with its disadvantages. Implicitly, an individual rating their job satisfaction will also be incorporating, at least to some extent, their personal assessment of their health in relation to their job. We have previously shown in HEAF [47] that job dissatisfaction was more likely in younger male workers and that the main perceptions of work that affected job dissatisfaction were lack of appreciation and/or a feeling of achievement, and difficulty with colleagues at work and/or feeling unfairly criticised. Job insecurity and dissatisfaction with pay were more likely to cause

dissatisfaction in the self-employed. Importantly however, our current findings suggest that if job satisfaction could be increased amongst older workers, this might enable longer working lives.

Our study had some limitations. First, full- or part-time working status was only ascertained at baseline so could not be incorporated in the longitudinal analysis file; this question has been reintroduced in recent follow-ups. Second, the information about HEAF participants is self-reported; face to face measures of physical capability and direct measurement of physical activity by wearable accelerometers would be valuable and a pilot study to assess the feasibility of this is underway. Third, we did not have a sufficient number of HRJL events to disaggregate analyses by permanent or temporary exits from employment, and moreover we could not be certain that exits that were not followed by a return to employment within only two years of follow-up would necessarily remain permanent. The ongoing annual longitudinal follow-ups of the HEAF cohort will generate valuable data on these older workers as they move through the retirement transition and we will be able to identify those who leave work permanently as opposed to those in whom there is a trajectory of work exits and re-entries. Fourth, we observed a bias towards healthier participants of higher socio-economic position in the sample of workers versus non-workers during the 2-year follow-up; a bias arising in most cohort studies [46]. Although the estimated incidence of HRJL might be lowered by this bias, the principal relationships that we have explored between participant characteristics and HRJL were internal to the sample. Our results should therefore only be biased if the relationships observed among responders were systematically different than among non-responders which seems unlikely. Fifth, our analyses were based on the sample of participants with complete information on the outcome and all explanatory variables used in each model (i.e. complete case analysis approach). We acknowledge that such an approach results in inefficient estimates and can lead to biased point estimates if the data are e.g. not missing at random. Yet, our analysis identified some strong associations between HRJL and many of the risk factors explored and the risk estimates presented in this study under complete case analysis models are plausible and in keeping with the results of previous studies, suggesting that if there is any effect of bias due to missingness, it is likely to be small. Future work will investigate the usefulness of multiple imputation techniques [48] in the HEAF study. Finally, we have not considered the interplay between specific morbidities and the characteristics identified as risk factors for HRJL in this analysis; however, this is an area for future work in HEAF given its linkage with the Clinical Practice Research Datalink at baseline.

Our study also has many strengths. First, HEAF is a contemporary, general population cohort of older working-age adults in England, widely geographically distributed, which strengthens the topicality and generalisability of our findings to the wider population. Second, HEAF is a large cohort study which has allowed us to explore sex differences in the employment characteristics of older workers. Third, the rich characterisation of HEAF cohort members enabled simultaneous investigation of a wide range of socio-demographic, lifestyle and employment characteristics as risk factors for HRJL; which few studies in the literature have achieved.

Whilst our findings begin to hint at the complexity of enabling work to older ages, there are some key messages for employers and policy-makers. It seems that employers would be well-advised to take a nuanced approach to retaining older workers, considering the needs of male and female workers separately and evaluating their perception of their physical and mental capability in relation to their assessment of the job demands. Attempts to measure, and improve, rates of job satisfaction could also lead to increased retention of older workers. Achieving this might, for example, involve consultation with older workers' representatives and implementing suitable and inclusive policies around flexible working to enable caring

responsibilities [43, 44]. For policy-makers, the message is clear that blanket policies of raising the age of eligibility for state pension and prohibiting age discrimination will not be universally effective at extending working lives: it is generally easier to remain in sedentary jobs than it is to stay working in those which are physically very demanding. Moreover, our results hint that the burden of HRJL among older workers will not be equal, and that those who are socioeconomically disadvantaged and struggling financially (for example as a consequence of the accumulated effects of lower pay, not owning their home, and a lack of private pension arrangements) will be more likely to need other governmental financial support through welfare benefits before they are old enough to claim their state pension. Going further however, what may be needed by policy-makers is a cultural shift from regarding capability to remain in work to older ages as a challenge for occupational health departments and employers to fix, to a recognition that a long working life is something that needs to be addressed and planned for by individuals, employers and public health policies throughout a person's life [49, 50]. In order to work later in life, mental and physical capacity needs to be preserved and these are both influenced by factors that operate throughout the lifecourse [51]. Our results that risk of HRJL is related to physical activity among men, and smoking and BMI among women, show that lifecourse health behaviours influence work ability beyond the age of 50 years. Employers and policy-makers could usefully consider ways in which to promote healthy behaviours throughout the lifecourse both at work and outside of the workplace, thereby benefitting all.

In summary, our results emphasise that the employment contexts and characteristics of contemporary older male and female workers are different and that characteristics such as job satisfaction and perceived ability to cope with the physical and mental demands of work are key determinants of HRJL. The implications are twofold; first for future research, and second for workplace interventions and policies aimed at extending working life. We recommend that future studies of older workers collect data longitudinally with time-varying covariates and consider sex differences, and investigate the predictive, confounding, or mediating role of a wide panel of potential risk factors for HRJL, with psychosocial work characteristics key amongst these. A next step will be to explore the interplay between the important factors identified in this study with specific morbidities also known to have a marked impact on premature exit from the workforce such as musculoskeletal disorders and mental health conditions. Our results also support calls for the development of policies and workplace interventions which: take a flexible, person-centred approach to extended working life [8, 52], recognise the importance of psychosocial work characteristics, and acknowledge heterogeneity between employees, occupations, and employment settings.

## Supporting information

**S1 Appendix. HEAF questions, response categories, coded analysis variables and reference categories (in italics).**
(PDF)

**S2 Appendix. Distribution of job exits between HEAF baseline and 2-year follow-up by sex and reason for leaving employment.**
(PDF)

**S3 Appendix.** (3a) and (3b) SOC2010 prevailing* 3-digit job code by sex and work pattern between HEAF baseline and 2-year follow-up.
(PDF)

**S1 File. HEAF consent form.**
(DOCX)

## Acknowledgments

We wish to thank the Clinical Practice Research Datalink and the 24 general practices that supported data collection; also, the staff of the MRC Lifecourse Epidemiology Unit who provided data entry and computing support (notably Vanessa Cox). Finally, we thank the HEAF participants for giving their time so generously to participate in the study.

## Author Contributions

**Conceptualization:** Holly E. Syddall, Karen Walker-Bone.

**Data curation:** Stefania D'Angelo, Georgia Ntani, Martin Stevens, E. Clare Harris, Catherine H. Linaker.

**Formal analysis:** Holly E. Syddall.

**Funding acquisition:** E. Clare Harris, Karen Walker-Bone.

**Methodology:** Holly E. Syddall, Georgia Ntani.

**Project administration:** Martin Stevens, E. Clare Harris, Catherine H. Linaker.

**Supervision:** E. Clare Harris, Karen Walker-Bone.

**Writing – original draft:** Holly E. Syddall.

**Writing – review & editing:** Holly E. Syddall, Stefania D'Angelo, Georgia Ntani, Martin Stevens, E. Clare Harris, Catherine H. Linaker, Karen Walker-Bone.

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
