## [Decision Letter · Decision Letter 0]

4 Jun 2020

PONE-D-20-01352

Work participation and risk factors for health-related job loss among older workers in the Health and Employment after Fifty (HEAF) study: evidence from a two year follow-up period

PLOS ONE

Dear Dr. Syddall,

Thank you for submitting your manuscript to PLOS ONE. After careful consideration, we feel that it has merit but does not fully meet PLOS ONE’s publication criteria as it currently stands. Therefore, we invite you to submit a revised version of the manuscript that addresses the points raised during the review process.

The manuscript touches upon interesting and timely aspects, but suffers froma number of weaknesses that have to be thoroughly and fully overcome before any further consideration can be given to it.

We look forward to receiving your revised manuscript.

Kind regards,

Denis Alves Coelho, PhD

Academic Editor

PLOS ONE

Journal Requirements:

2. Please provide additional details regarding participant consent. In the ethics statement in the Methods and online submission information, please ensure that you have specified (a) whether consent was informed and (b) what type you obtained (for instance, written or verbal, and if verbal, how it was documented and witnessed). If your study included minors, state whether you obtained consent from parents or guardians. If the need for consent was waived by the ethics committee, please include this information.

Additional Editor Comments (if provided):

The reviewers identified several critical shortcomings in the manuscript, besides some interesting strengths. That said, this invitation to revise the manuscript is a single opportunity to fix all the problems identified in both reviews.

Reviewers' comments:

Reviewer's Responses to Questions

**Comments to the Author**

1. Is the manuscript technically sound, and do the data support the conclusions?

Reviewer #1: Yes

Reviewer #2: No

2. Has the statistical analysis been performed appropriately and rigorously? 

Reviewer #1: Yes

Reviewer #2: No

3. Have the authors made all data underlying the findings in their manuscript fully available?

Reviewer #1: Yes

Reviewer #2: Yes

4. Is the manuscript presented in an intelligible fashion and written in standard English?

Reviewer #1: Yes

Reviewer #2: Yes

5. Review Comments to the Author

Reviewer #1: The manuscript, “Work participation and risk factors for health-related job loss among older workers in the Health and Employment after Fifty (HEAF) study: evidence from a two year follow-up period” investigates risk factors for health-related job loss (HRJL) among a cohort study of older adults in England, aged 50-64 years at baseline. In the opinion of this reviewer, the manuscript has a potential to make a contribution to the field of public aging and health, but concerns (see below) must first be addressed. This review focuses on conceptualization of approach and communication of key findings.

Abstract:

1. Because older age thresholds generally begin at age 65 (and the included age group of the study is 50-64), this reviewer recommends changing “older adults” to “middle aged and older adults” or simply “adults aged 50-64 (page 2, line 46 and throughout manuscript).

Introduction:

2. Avoid beginning paragraphs with “however” and “moreover” (page 4, lines 74, 84).

3. At what point might physical or mental strain have deleterious consequences on health? Are individual protected from age 20-40 and unprotected in later life? This reviewer recommends bringing in outside citations and removing the terms “is unlikely to be exactly the same in later life”, which seems assumptive and somewhat vague. It’s also important to note the many potential positive impacts of work on physical, mental, cognitive, and social health and well-being. Many individuals thrive in work later into their 60s, 70s, and 80s and that should also be recognized here. Also, beyond the “dependency ratio”, older worker have a great deal to contribute. The authors might consider acknowledging how older workers increase the diversity of the workforce and also increasingly reflect the consumer base.

Methods:

4. What is the rationale for choosing the examined criteria over others? It would be helpful if these factors were explained in a more structured or cohesive way, ideally with a theoretical model, with relevant literature on prior findings regarding the relationship between these factors and HRJL. There are many covariates and the rationale for inclusion should be more clearly stated – a more focused analysis would greatly improve the readability of this paper and its tables (Descriptive tables should fit on one page). Also, was it possible to examine chronic illness co-morbidity or disability/functional limitations?

5. The statistical methods should be available for review in the main body of the manuscript and not in an appendix.

6. The measures of “obese” and “not obese” are less relevant as individuals age, when frailty is also a concern. This reviewer recommends re-categorizing into “underweight” “about the right weight” “overweight” and “obese” (or a minimum of 3 groupings) for improved analyses (Appendix 2)

7. If the data allow, this reviewer suggests making smoking status into three categories (never/previous/current) rather than two categories (never/ex or current) to improve analysis (Appendix 2).

8. This reviewer is confused by the statement, “potential responder bias was examined by cross-tabulations and chi-squared tests of differences in the baseline characteristics”. Perhaps the authors intended to refer to potential recall error? (Appendix 2)

9. In the methods, you refer to “Cox proportional hazards models”, while the approach is identified as Cox’s proportional hazards models in the abstract. Either are correct but it’s important to be consistent throughout the manuscript (page 6, line 153).

10. What was the rationale for using this step-wise approach over a more parsimonious strategy? (page 6, line 151-161).

11. How were missing data handled?

Results:

12. The sampling strategy should be included in the Methods section, rather than the Results section (page 6, lines 165-168).

13. This reviewer is confused how “mortgaging rather than owning their home outright” is evidence of a “healthy worker effect” (page 7, line 176).

14. This reviewer recommends omitting or greatly revising Figure 1, which contains more information than readers are likely able to process.

Discussion/Limitations:

15. It is important to note that – due to the 2-year follow up only – many of these exits are likely not permanent and that a trajectory analysis would be helpful in future studies to capture these exits and potential re-entries. It would be helpful to compare the prevalence of job exit among this sample with job exit in younger samples (page 17, lines 300-311).

16. This reviewer appreciated this discussion of policy implications on gendered findings. It is also important to note that many consider disability to relate to a “mismatch between a person and their environments”, which could include working environments. How could working environments be improved to maintain older workers, and to allow them to thrive? (page 17, lines 322 – page 18, lines 330).

17. The fact that the socioeconomic variables were no longer significant in the adjusted models does not necessarily suggest that proximity to retirement, physical activity, job satisfaction, and difficulty coping with the demands at work operated as mediators (page 18, lines 335-339). A mediation analysis is needed to make this claim and the model could very well be under-powered.

18. The discussion preceding the statement, “Another way in which social inequalities might affect risk of HRJL” (page 19, line 734) concerns differential health behaviors by socioeconomic position. This reviewer recommends either (a) discussing more about the fundamental causes that could underlie those differential health behaviors, linking to inequalities or (b) changing “inequalities” to “factors”.

19. In what direction do the authors anticipate the healthy participation effect might have influenced the results? (page 20, lines 370-372).

20. Please include a citation to the study (page 20, lines 394-397).

21. In what direction might the estimated incidence of HRJL be affected by the healthy participants bias? (page 20, lines 408-409).

22. Further discussion of the handling of missing data would be helpful here (page 20, lines 409-410).

Reviewer #2: Revision Article

-The article PONE-D-20-01352 aimed to explore risk factors for health-related job loss over 2 years of follow-up, among a cohort of contemporary older workers in England. To this purpose the authors analyzed labor-force transitions of 5,032 individuals aged 50-64 (at baseline), using two rounds of the Health and Employment after Fifty (HEAF) panel study, and employing semi-parametric survival analysis.

-The study topic is appealing, timely and relevant, given the focus on a relevant life transition in ageing societies. However, after reading the manuscript twice I have multiple major concerns regarding the introduction of the study, the discussion of previous literature, and the novelty of this article, which if they are not seriously considered, I would suggest to the Editor of PLOS ONE not to publish this manuscript.

Introduction

-The most notorious weakness of the introduction is that it does not identify appropriately a knowledge gap in this study field that needs to be addressed. Author actually mention that a “focus on identification of the wider drivers of health-related job loss, irrespective of specific illnesses or diagnoses, therefore offers the potential to inform the development and implementation of workplace strategies to encourage and enable work to older ages.” However, this is not really original in the field of retirement and health studies. So, from my point of view, it is not clear why the main study subject (i.e., risk factors for health-related job loss) deserves to be explored, challenged, or revisited. I am not saying this is not an important subject, but rather that its importance is not clearly discussed or explained to the reader.

-Also, the discussion held in the introduction is not really coherent. Specifically, authors provide first a discussion on extended working policies to then mention that health is a major determinant for early labor market exits. Given that the allowed number of words is limited, if the main topic of this research is the determinants of health-related job loss (which leads individuals to early exit from the labor market), I would expect a more focused theoretical discussion on early retirement due to health reasons, and particularly on what remains unexplored in this field.

Data & Methods

-There is no justification provided about why women and men are separated in the analysis.

-There is no test of the main assumption of the statistical models used in this research: proportional hazard of predictors over the dependent variable.

Discussion

-I believe authors once again did not express clearly what the novelty or originality of this study is. Authors for instance mention: “Our work is novel in considering self-reported exit from employment mainly or partly for health reasons”; but this is only partly true. Then they state:

“Our data are novel in pertaining to a contemporary cohort of older workers who are progressing through the retirement transition at a time when recent legislation to raise and harmonise state pension age is taking effect”; however, I am highly doubtful on whether the recent modifications in the retirement legislation affected the cohorts studied in this research.

-Finally, from my perspective, the discussion about policy contributions of this research does not provide any original idea to the debate on extended working lives. Previous retirement research has provided very similar reflections during the last 10-12 years.

-More importantly, in this section I expected an in-depth reflection on how the study findings might help scholars as well as also policy makers to approach differently to the understanding of the closed relationship between work, health, retirement and sociodemographic factors (such as genders, educational levels, or cohorts). But this is unfortunately not achieved.

6. PLOS authors have the option to publish the peer review history of their article (what does this mean?). If published, this will include your full peer review and any attached files.

Reviewer #1: Yes: Emily Joy Nicklett

Reviewer #2: No

---

## [Author Response · Author response to Decision Letter 0]

10 Aug 2020

Please see uploaded documents: 'PLOS One covering letter_revision' and 'Response to reviewers'.

---

## [Decision Letter · Decision Letter 1]

7 Sep 2020

Work participation and risk factors for health-related job loss among older workers in the Health and Employment after Fifty (HEAF) study: evidence from a 2-year follow-up period

PONE-D-20-01352R1

Dear Dr. Syddall,

We’re pleased to inform you that your manuscript has been judged scientifically suitable for publication and will be formally accepted for publication once it meets all outstanding technical requirements.

Kind regards,

Denis Alves Coelho, PhD

Academic Editor

PLOS ONE

Reviewers' comments:

Reviewer's Responses to Questions

**Comments to the Author**

1. If the authors have adequately addressed your comments raised in a previous round of review and you feel that this manuscript is now acceptable for publication, you may indicate that here to bypass the “Comments to the Author” section, enter your conflict of interest statement in the “Confidential to Editor” section, and submit your "Accept" recommendation.

Reviewer #2: All comments have been addressed

2. Is the manuscript technically sound, and do the data support the conclusions?

Reviewer #2: Partly

3. Has the statistical analysis been performed appropriately and rigorously? 

Reviewer #2: Yes

4. Have the authors made all data underlying the findings in their manuscript fully available?

Reviewer #2: Yes

5. Is the manuscript presented in an intelligible fashion and written in standard English?

Reviewer #2: Yes

6. Review Comments to the Author

Reviewer #2: I feel that authors in this new version have addressed all my suggestions and comments. So I would suggest to Editor of PLOS ONE to publish the manuscript.

7. PLOS authors have the option to publish the peer review history of their article (what does this mean?). If published, this will include your full peer review and any attached files.

Reviewer #2: No

---

## [Editor Report · Acceptance letter]

10 Sep 2020

PONE-D-20-01352R1 

Work participation and risk factors for health-related job loss among older workers in the Health and Employment after Fifty (HEAF) study:evidence from a 2-year follow-up period 

Dear Dr. Syddall:

I'm pleased to inform you that your manuscript has been deemed suitable for publication in PLOS ONE. Congratulations! Your manuscript is now with our production department. 

Kind regards, 

on behalf of

Dr. Denis Alves Coelho 

Academic Editor

PLOS ONE